# Adequacy of Pain Management in Patients Referred for Radiation Therapy: A Subanalysis of the Multicenter ARISE-1 Study

**DOI:** 10.3390/cancers16010109

**Published:** 2023-12-25

**Authors:** Costanza M. Donati, Chiara Maria Maggiore, Marco Maltoni, Romina Rossi, Elena Nardi, Alice Zamagni, Giambattista Siepe, Filippo Mammini, Francesco Cellini, Alessia Di Rito, Maurizio Portaluri, Cristina De Tommaso, Anna Santacaterina, Consuelo Tamburella, Rossella Di Franco, Salvatore Parisi, Sabrina Cossa, Vincenzo Fusco, Antonella Bianculli, Pierpaolo Ziccarelli, Luigi Ziccarelli, Domenico Genovesi, Luciana Caravatta, Francesco Deodato, Gabriella Macchia, Francesco Fiorica, Giuseppe Napoli, Milly Buwenge, Alessio G. Morganti

**Affiliations:** 1Radiation Oncology Unit, IRCCS Azienda Ospedaliero-Universitaria di Bologna, 40138 Bologna, Italy; giambattista.siepe@aosp.bo.it (G.S.); filippo.mammini@studio.unibo.it (F.M.); alessio.morganti2@unibo.it (A.G.M.); 2Department of Medical and Surgical Sciences (DIMEC), Alma Mater Studiorum University of Bologna, 40138 Bologna, Italy; cmmaggiore@gmail.com (C.M.M.); marcocesare.maltoni@unibo.it (M.M.); romina.rossi@irst.emr.it (R.R.); alice.zamagni@yahoo.it (A.Z.); mbuwenge@gmail.com (M.B.); 3Medical Oncology Unit, IRCCS Azienda Ospedaliero-Universitaria di Bologna, 40138 Bologna, Italy; 4Palliative Care Unit, AUSL Romagna, 40121 Forlì, Italy; 5Cardiology Unit, Cardiac Thoracic and Vascular Department, IRCCS Azienda Ospedaliero-Universitaria di Bologna, 40138 Bologna, Italy; elena.nardi2@unibo.it; 6Department of Experimental, Diagnostic and Specialty Medicine, University of Bologna, 40126 Bologna, Italy; 7Dipartimento di Scienze Radiologiche, Radioterapiche ed Ematologiche, IRCCS, UOC di Radioterapia, Fondazione Policlinico Universitario A. Gemelli, 00168 Roma, Italy; francesco.cellini@policlinicogemelli.it; 8Istituto di Radiologia, Università Cattolica del Sacro Cuore, 00168 Roma, Italy; 9IRCCS Istituto Tumori “Giovanni Paolo II”, 70124 Bari, Italy; aledirito@yahoo.it; 10General Hospital “Perrino”, 72100 Brindisi, Italy; portaluri@hotmail.com (M.P.); detommasocristina@gmail.com (C.D.T.); 11U.O. di Radioterapia AOOR PAPARDO PIEMONTE, 98121 Messina, Italy; anna.santacaterina@virgilio.it (A.S.); consu.universitaly@gmail.com (C.T.); 12S.C. di Radioterapia dell’Istituto Nazionale Tumori Pascale, 80131 Napoli, Italy; r.difranco@istitutotumori.na.it; 13Radioterapia Opera di S. Pio da Pietralcina, Casa Sollievo della Sofferenza, 71013 San Giovanni Rotondo, Italy; s.parisi@operapadrepio.it (S.P.); s.cossa@operapadrepio.it (S.C.); 14IRCCS CROB, 85028 Rionero in Vulture, Italy; fuscovincenzo@hotmail.com (V.F.); antonellabianculli@gmail.com (A.B.); 15U.O. Radioterapia Oncologica, S.O. Mariano Santo, 87100 Cosenza, Italy; pziccarelli@virgilio.it (P.Z.); lziccarelli@virgilio.it (L.Z.); 16Radiation Oncology Unit, Università degli Studi G. D’Annunzio, 66100 Chieti, Italy; d.genovesi@unich.it (D.G.); lcaravatta@hotmail.com (L.C.); 17Radiotherapy Unit, Gemelli Molise Hospital, Catholic University of Sacred Heart, 86100 Campobasso, Italy; francesco.deodato@responsible.hospital (F.D.); macchiagabriella@gmail.com (G.M.); 18U.O.C. di Radioterapia e Medicina Nucleare, Ospedale Mater Salutis di Legnago, 37045 Verona, Italy; francesco.fiorica@aulss9.veneto.it (F.F.); napoligiuseppe.84@gmail.com (G.N.)

**Keywords:** pain management, cancer patients, radiotherapy, pain management index, modified pain management index, palliative care, multidisciplinary care

## Abstract

**Simple Summary:**

Cancer patients frequently experience pain, impacting their quality of life. Unfortunately, pain management in those referred for radiotherapy (RT) is often insufficient, with limited research in this area. This study aimed to assess the adequacy and effectiveness of pain management and identify factors affecting them in cancer patients referred for RT. We observed 1042 cancer outpatients and found that 42.9% did not receive adequate pain management. Specifically, 72% of patients referred for palliative RT and 75% of those referred for curative RT experienced inadequate or ineffective analgesic therapy. Patients undergoing palliative RT, those with poorer general health, those with cancer-related pain, and those treated in Northern Italy had higher odds of receiving adequate pain management. Our findings highlight the need for educational and organizational strategies to address this issue and suggest that early palliative RT referral can improve pain management and treatment outcomes for cancer patients.

**Abstract:**

Background: Pain is a prevalent symptom among cancer patients, and its management is crucial for improving their quality of life. However, pain management in cancer patients referred to radiotherapy (RT) departments is often inadequate, and limited research has been conducted on this specific population. This study aimed to assess the adequacy and effectiveness of pain management when patients are referred for RT. Moreover, we explored potential predictors of adequate pain management. Methods: This observational, prospective, multicenter cohort study included cancer patients aged 18 years or older who were referred to RT departments. A pain management assessment was conducted using the Pain Management Index (PMI), calculated by subtracting the pain score from the analgesic score (PMI < 0 indicated inadequate pain management). Univariate and multivariate analyses were performed to identify predictors of adequate pain management. Results: A total of 1042 cancer outpatients were included in the study. The analysis revealed that 42.9% of patients with pain did not receive adequate pain management based on PMI values. Among patients with pain or taking analgesics and referred to palliative or curative RT, 72% and 75% had inadequate or ineffective analgesic therapy, respectively. The odds of receiving adequate pain management (PMI ≥ 0) were higher in patients undergoing palliative RT (OR 2.52; *p* < 0.001), with worse ECOG-PS scores of 2, 3 and 4 (OR 1.63, 2.23, 5.31, respectively; *p*: 0.017, 0.002, 0.009, respectively) compared to a score of 1 for those with cancer-related pain (OR 0.38; *p* < 0.001), and treated in northern Italy compared to central and southern of Italy (OR 0.25, 0.42, respectively; *p* < 0.001). Conclusions: In this study, a substantial proportion of cancer patients referred to RT departments did not receive adequate pain management. Educational and organizational strategies are necessary to address the inadequate pain management observed in this population. Moreover, increasing the attention paid to non-cancer pain and an earlier referral of patients for palliative RT in the course of the disease may improve pain response and treatment outcomes.

## 1. Introduction

Pain is a pervasive and distressing symptom experienced by a significant number of cancer patients. In fact, the National Cancer Institute has recognized pain, along with depression and fatigue, as one of the “priority symptoms” experienced by cancer patients that needs comprehensive assessment and treatment [1]. Despite this recognition, pain management in cancer patients remains a significant clinical challenge, with a substantial proportion of patients receiving inadequate pain control.

While there have been efforts to improve pain management in various clinical settings, including medical oncology and palliative care, relatively little attention has been given to pain management specifically in patients referred to radiotherapy (RT) departments [2,3,4]. RT plays a crucial role in cancer treatment, aiming to deliver targeted doses of radiation to cancerous tissues while minimizing damage to surrounding healthy cells. However, the potential for pain and discomfort during and after RT is a well-recognized concern.

To address this gap in knowledge and improve pain management in patients referred for RT, we conducted a subanalysis of a multicenter observational study known as the ARISE-1 study [5]. The primary objective of the ARISE-1 trial was to evaluate the overall adequacy of pain management in patients treated within RT departments. In this subanalysis, we aimed to specifically assess the adequacy of pain management in the sub-population of patients who were evaluated in RT centers before the delivery of treatment, specifically at their initial visit. In this manner, we intended to assess the referral patterns of physicians who seek an RT evaluation for their patients.

Additionally, we sought to explore the association between the adequacy of pain management, as measured by the Pain Management Index (PMI), and various pain characteristics and other potentially predictive factors. By examining these associations, we aimed to gain a comprehensive understanding of the factors influencing pain management outcomes in patients undergoing RT.

The findings from this subanalysis have the potential to inform clinical practice and highlight areas where improvements in pain management can be made. Ultimately, our goal is to enhance the overall quality of care and ensure optimal pain control for cancer patients undergoing RT.

## 2. Materials and Methods

### 2.1. Study Design

The ARISE study was an observational, prospective, multicenter cohort study designed to assess pain management in cancer patients referred to RT departments. The study received ethical approval from the ethics committees of the participating centers (ARISE 327/2017/O/Oss). Patients enrolled in the study provided written informed consent. For the purpose of this analysis, we selected patients who underwent a pain management assessment during their first visit to the RT department. All patients who met the inclusion criteria and underwent the initial medical examination visit at the participating centers between October and November 2019 were included.

### 2.2. Inclusion and Exclusion Criteria

The inclusion criteria for this analysis were as follows: patients with cancer (regardless of the primary tumor, the tumor stage, and the aim of RT), patients referred to RT departments, and patients aged 18 years or older. Patients with comorbidities (e.g., psychiatric disorders or neurosensory deficits) that could hinder data collection or prevent the granting of consent were excluded from the analysis.

### 2.3. Data Collection

Data were collected using a structured form during the patient’s visit. The recorded information included gender, age, the Eastern Cooperative Oncology Group Performance Status Scale (ECOG-PS) [6], the aim of RT, the primary cancer, the tumor stage, the type of pain (cancer pain, non-cancer pain, mixed pain), the intensity of pain measured with the Numerical Rating Scale (NRS) [7,8], and an analgesic score.

### 2.4. End Points

Pain intensity was categorized using the NRS as follows: 0 (no pain), 1 (mild pain; NRS: 1–4), 2 (moderate pain; NRS: 5–6), and 3 (severe pain; NRS: 7–10). An analgesic score was assigned based on the therapy received by the patients: 0 (no pain medication), 1 (non-opioid analgesics), 2 (use of “weak” opioids), and 3 (use of “strong” opioids). The therapy was previously established by the referring clinician, according to the guidelines, and was not changed during the study [9,10,11].

To assess the adequacy of pain management, the Pain Management Index (PMI) was calculated by subtracting the pain score from the analgesic score. A negative PMI value indicated inadequate analgesic prescription [12]. However, the PMI is known to have limitations [13]. Notably, patients experiencing intense pain despite strong opioid medication may still have a PMI value ≥ 0, inaccurately suggesting adequate pain management. To address this discrepancy, our study introduces a secondary analysis that classifies patients into two distinct groups: Group A consists of patients with PMI < 0 or those with PMI ≥ 0 but experiencing substantial pain (NRS > 4), indicative of poorly managed or ineffective pain therapy; Group B includes patients with PMI ≥ 0 and a pain score ≤ 4, representing those with adequate and effective pain management.

### 2.5. Statistical Analysis

To identify potential predictors of adequate pain management, the following parameters were explored: gender, age, the aim of RT, the ECOG-PS, the primary tumor, the stage of disease, the type of pain, and the location of the RT center. The chi-square test was used to assess the statistical significance of associations, with a *p*-value < 0.05 considered significant. All variables from the univariate analysis were included in the multivariable logistic regression analysis to confirm potential predictors of adequate PMI. The statistical analysis was performed using SYSTAT, version 11.0 (SPSS, Chicago, IL, USA).

## 3. Results

### 3.1. Patient Characteristics

A total of 1042 cancer outpatients aged 18 years or older were included in this sub-analysis. Among them, 48.8% received RT with curative intent, while 51.2% received RT with palliative intent. The tumor stage was non-metastatic in 57.4% of patients and metastatic in 42.6%. Pain was reported by 757 patients, with 40.5% experiencing cancer-related pain, 17.3% having non-cancer pain, and 14.9% having mixed pain. Among the patients, 608 were taking analgesic drugs. Detailed patient characteristics are presented in Table 1.

#### 3.1.1. Pain Management Index (PMI)

Considering all patients enrolled in the study, the rate of subjects with PMI < 0 was 31.2% (Figure 1).When only considering patients with pain or receiving analgesics, the rate of PMI < 0 increased to 42.9% (Figure 2). When focusing on patients referred for RT with a palliative intent, 28.3% had PMI < 0 (Figure 3). Among patients with pain or taking analgesics in this group, 30.6% had PMI < 0 (Figure 4). For patients referred for RT with a curative intent, 34.3% had PMI < 0 (Figure 5). Among patients with pain or taking analgesics in this group, the proportion with PMI < 0 was even higher at 66.2% (Figure 6).

#### 3.1.2. Analysis of Patients with Inadequately or Ineffectively Managed Pain

Considering all patients enrolled in the study, the rate of patients in Group A (with PMI < 0 or with PMI ≥ 0 but experiencing substantial pain: NRS > 4) receiving poorly managed or ineffective pain therapy was 53% (Figure 1). When considering only patients with pain or taking analgesics, the rate of patients in the Group A increased to 73% (Figure 2). Moreover, the rate of patients in Group A was 66% (Figure 3), 72% (Figure 4), 39% (Figure 5) and 75% (Figure 6) in patients referred for palliative RT, in the latter with pain or taking analgesics, in patients referred for curative RT, and in the latter with pain or taking analgesics, respectively.

#### 3.1.3. Predictors of Pain Management Adequacy

The univariate analysis identified several parameters significantly correlated with PMI < 0, as shown in Table 2. The multivariate analysis performed in the same patient population confirmed the following predictors of adequate pain management or PMI ≥ 0 (Table 3):Aim of treatment: Compared to patients undergoing RT with curative intent (reference category), patients undergoing RT with a palliative intent had higher odds of receiving adequate pain management (PMI ≥ 0, OR 2.52; *p*-value < 0.001).ECOG Performance Status: Compared to patients with an ECOG-PS score of 1 (reference category), patients with ECOG-PS scores of 2, 3, and 4 had higher odds of receiving adequate pain management (PMI ≥ 0, OR 1.63, 2.23, 5.31, respectively; *p*-value 0.017, 0.002, 0.009, respectively).Type of pain: Compared to those with cancer-related pain (reference category), patients with non-cancer pain had lower odds of receiving adequate pain management (PMI ≥ 0, OR 0.38; *p*-value < 0.001).Location of the RT center: Compared to patients treated in RT departments in northern Italy (reference category), patients treated in the center and south of Italy had lower odds of receiving adequate pain management (PMI ≥ 0, OR 0.25, 0.42, respectively; *p*-value < 0.001).

## 4. Discussion

In this multicenter study including over one thousand cancer patients referred to RT departments and evaluated during their first visit, we found that 42.9% of patients with pain did not receive adequate pain management (PMI < 0). Moreover, the rate of patients with pain inadequately or ineffectively managed (Group A: PMI < 0 or PMI ≥ 0 but NRS > 4) was 73%.

Our findings underscore the need for improved pain management in patients referred to RT departments. One potential strategy to enhance pain control is to refer patients for palliative RT at earlier stages of the disease. Several studies have attempted to identify predictors of pain relief after RT for painful tumors [14,15]. It has been observed that patients with a shorter duration of pre-RT pain have a higher incidence of pain response to treatment, specifically in terms of pain caused by the irradiated tumors themselves [16,17,18]. The influence of pain duration can be attributed to the fact that acute pain often responds well to analgesic therapies, whereas chronic pain is more challenging to manage [19,20].

To date, there is no consensus on the optimal timing for referring cancer patients for palliative RT. However, many studies have demonstrated that initiating palliative treatment at earlier stages of the disease leads to less aggressive care, more adequate pain management, an improved quality of life, and increased survival rates [13,21,22,23]. Moreover, referring patients to palliative RT earlier may increase the probability of achieving a better pain response and overall treatment outcomes [24].

Furthermore, our analysis revealed that patients with better clinical conditions, such as a good performance status and non-neoplastic pain, had a higher incidence of inadequate pain management. This finding is consistent with previous studies [25,26]. In a review of cancer pain undertreatment, Deandrea et al. found that patients with a good performance status are often inadequately treated [27]. This may be attributed to the fact that patients who appear clinically less ill may be perceived to have lower pain scores and thus considered to require less potent analgesics. These observations suggest that a failure in physician–patient communication and an overreliance on performance status as an indicator of pain intensity may contribute to undertreatment. To address this issue, pain management should be dictated by a patient’s pain score rather than making a subjective assessment based solely on performance status.

Moreover, our subanalysis revealed that patients with non-neoplastic pain, resulting from benign comorbidities, were more likely to have inadequate pain management. This finding is consistent with other studies [28,29] and highlights the importance of recognizing and adequately managing pain in patients with non-cancer-related conditions [30].

Interestingly, our analysis also showed geographic variations in the adequacy of pain management within Italy. Patients treated in the center and south of Italy had a higher incidence of inadequate pain management compared to those treated in northern Italy. Similar geographic variations in the adequacy of analgesic therapy have been reported in studies conducted in other regions [29]. These variations may reflect differences in healthcare practices, resource allocation, or cultural factors. Further investigation is needed to better understand these geographic disparities and implement interventions to ensure equitable pain management across regions.

Moreover, in reflecting upon the scope of our study, we acknowledge certain limitations in the dataset utilized. Notably, our analysis did not include factors such as BMI, the presence of cancer markers, the metastatic site, and a history of smoking and alcohol consumption. These elements have a potential impact on the efficacy of pain management in radiotherapy patients. This limitation stems from the practical constraints encountered in the multicenter ARISE-1 study, our primary data source, which involved an extensive cohort of over 2000 patients. As such, our findings should be interpreted with consideration of these omitted variables, and future research may benefit from their inclusion to further elucidate the complexities of pain management in this context.

It is worth noting that the overall rate of patients referred to RT departments who did not receive adequate analgesic therapy deserves attention, as it nears 50%. To address this issue, a systematic registration of the Pain Management Index (PMI) alongside pain assessment could serve as a screening tool to identify patients with inadequate pain management. Moreover, educational strategies for medical and nursing staff should be implemented to improve awareness of pain management and enhance the ability to identify and treat patients with painful symptoms. Additionally, multidisciplinary collaborations, such as the involvement of multidisciplinary teams or joint clinics, can contribute to improved symptom management in RT departments [31].

In summary, our study reveals a significant percentage of cancer patients referred to RT departments who did not receive adequate pain management. This finding appears to be attributed to the significant oversight of non-cancer pain in patients referred for curative RT and, in most instances, the decision to opt for palliative RT when analgesic therapy is no longer effective. Referring patients for palliative RT earlier may improve the probability of better pain responses and treatment outcomes. Educational and organizational strategies are needed to reduce the proportion of patients with inadequate pain management. By addressing these issues, we can strive to provide optimal pain control and enhance the overall quality of care for cancer patients in RT settings.

## 5. Conclusions

Our analysis highlights that pain management was inadequate in 42.9% of cancer patients experiencing pain who were referred to Italian RT departments. Initiating palliative RT earlier in the course of the disease may improve pain response and overall treatment outcomes. Educational and organizational strategies are necessary to reduce the non-negligible percentage of patients with inadequate pain management and enhance the quality of care provided to cancer patients.

## Figures and Tables

**Figure 1 cancers-16-00109-f001:**
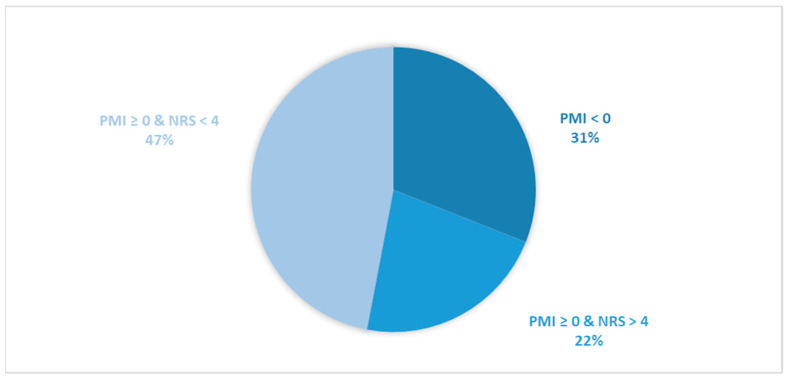
Pie chart depicting the distribution of patients based on the Pain Management Index (PMI) and Numerical Rating Scale (NRS) scores. All patients enrolled in the study (*n* = 1042) were included in the analysis.

**Figure 2 cancers-16-00109-f002:**
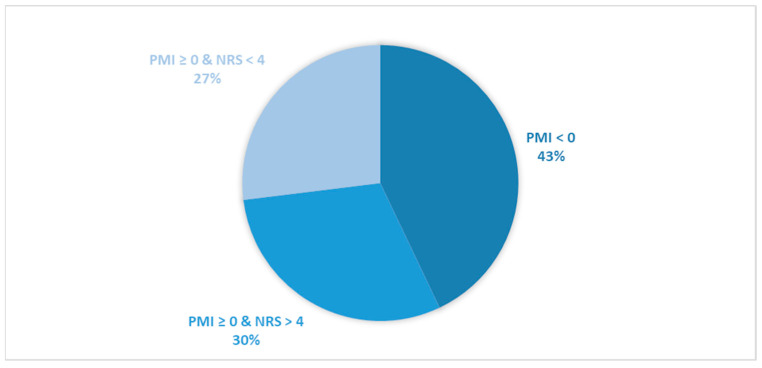
Pie chart illustrating the distribution of patients with pain or receiving analgesics based on the Pain Management Index (PMI) and Numerical Rating Scale (NRS) scores. Only patients with pain or receiving analgesics were included in the analysis (*n =* 757).

**Figure 3 cancers-16-00109-f003:**
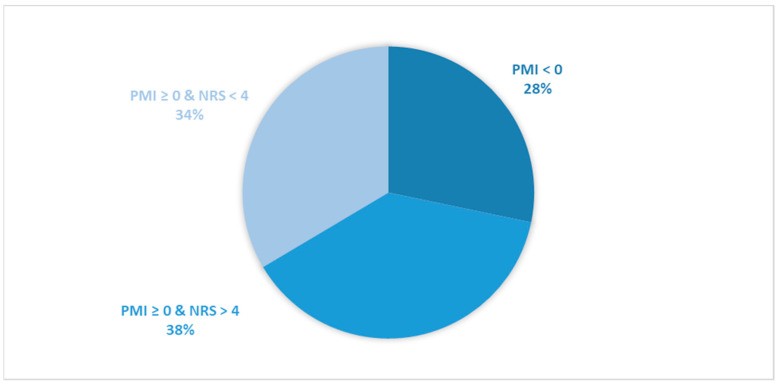
Pie chart presenting the distribution of patients based on the Pain Management Index (PMI) and Numerical Rating Scale (NRS) scores. Only patients undergoing palliative radiotherapy were included in the analysis (*n =* 534).

**Figure 4 cancers-16-00109-f004:**
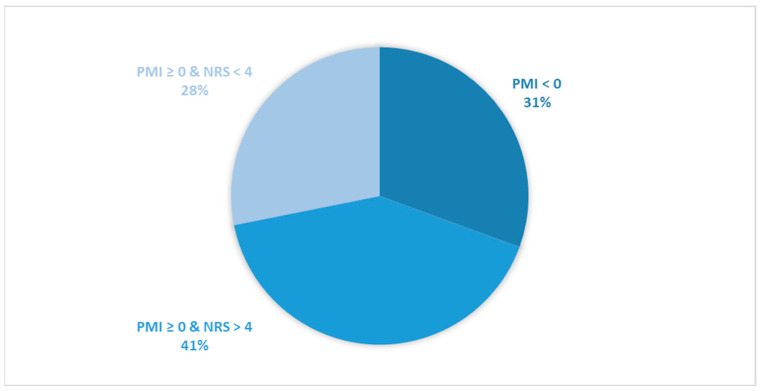
Pie chart displaying the distribution of patients based on the Pain Management Index (PMI) and Numerical Rating Scale (NRS) scores. Only patients undergoing palliative radiotherapy and with pain or receiving analgesics were included in the analysis (*n =* 494).

**Figure 5 cancers-16-00109-f005:**
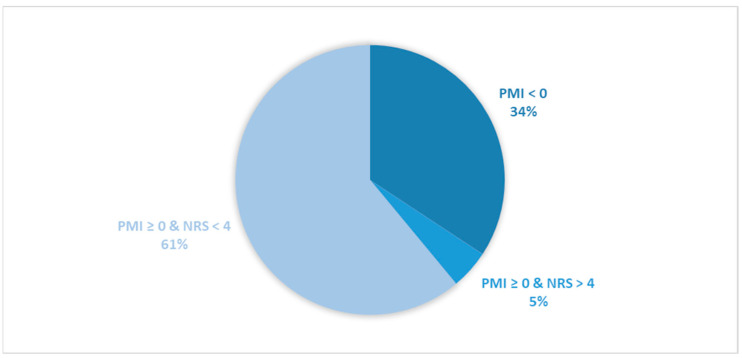
Pie chart illustrating the distribution of patients based on the Pain Management Index (PMI) and Numerical Rating Scale (NRS) scores. Only patients undergoing curative radiotherapy were included in the analysis (*n =* 508).

**Figure 6 cancers-16-00109-f006:**
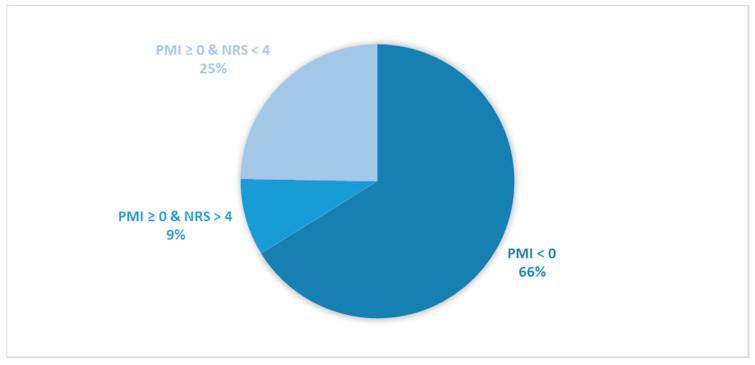
Pie chart presenting the distribution of patients based on the Pain Management Index (PMI) and Numerical Rating Scale (NRS) scores. Only patients undergoing curative radiotherapy and with pain or receiving analgesics were included in the analysis (*n =* 263).

**Table 1 cancers-16-00109-t001:** Patients’ Characteristics.

		Number	(%)
Gender			
Male		488	46.8
Female		554	53.2
Age, years			
≤70		659	63.2
71–80		279	26.8
>80		104	10.0
ECOG-PS			
0		226	21.7
1		461	44.2
2		208	20.0
3		122	11.7
4		25	2.4
Aim of treatment			
Curative		508	48.8
Palliative		534	51.2
Primary tumor			
Breast		333	32.0
Prostate		140	13.4
Gastrointestinal		102	9.8
Endometrial/cervical		52	5.0
Lung		143	13.7
Head and neck		63	6.0
Others		209	20.1
Tumor stage			
Metastatic		444	42.6
Non-metastatic		598	57.4
Type of pain			
Cancer pain		422	40.5
Non-cancer pain		180	17.3
Mixed pain		155	14.9
Pain score			
(NRS: 0)	0	314	30.1
(NRS: 1–4)	1	274	26.3
(NRS: 5–6)	2	304	29.2
(NRS: 7–10)	3	150	14.4
Analgesic score			
(No therapy)	0	434	41.7
(Analgesics)	1	299	28.7
(Weak opioids)	2	123	11.8
(Strong opioids)	3	186	17.9
Location of the radiotherapy center			
North of Italy		251	24.1
Center of Italy		132	12.7
South of Italy		659	63.2

Legend: ECOG-PS: Eastern Cooperative Oncology Group Performance Status Scale; NRS: Numerical Rating Scale.

**Table 2 cancers-16-00109-t002:** Univariate analysis on Pain Management Index: odds ratios of adequate pain management (only 757 patients with pain or undergoing analgesic therapy included).

				PMI					
		All Patients		<0		≥0			
				n	%	n	%		
All Patients				325	42.9	432	57.1	OR	*p*-Value
Gender	Male	380	50.2	147	38.7	233	61.3	Ref.	
	Female	377	49.8	178	47.2	199	52.8	0.705	0.017
Age, years	≤70	470	62.1	195	41.5	275	58.5	Ref.	
	71–80	198	26.2	95	48.0	103	52.0	0.768	0.122
	>80	89	11.8	35	39.3	54	60.7	1.094	0.703
Treatment aim	Curative	263	34.7	174	66.2	89	33.8	Ref.	
	Palliative	494	65.3	151	30.6	343	69.4	4.44	<0.001
ECOG-PS	1	415	54.8	224	54.0	191	46.0	Ref.	
	2	201	26.6	68	33.8	133	66.2	2.40	<0.001
	3	117	15.5	30	25.6	87	74.4	3.56	<0.001
	4	24	3.2	3	12.5	21	87.5	8.61	<0.001
Primary tumor	Breast	211	27.9	116	55.0	95	45.0	Ref.	
	Prostate	81	10.7	36	44.4	45	55.6	1.52	0.107
	Gastrointestinal	72	9.5	26	36.1	46	63.9	2.16	0.006
	Endometrial/cervical	33	4.4	19	57.6	14	42.4	0.89	0.78
	Lung	124	16.4	33	26.6	91	73.4	3.36	<0.001
	Head and neck	55	7.3	32	58.2	23	41.8	0.87	0.67
	Others	181	23.9	63	34.8	118	65.2	2.28	<0.001
Type of pain	Cancer pain	422	55.7	135	32.0	287	68.0	Ref.	
	Non-cancer pain	180	23.8	135	75.0	45	25.0	0.15	<0.001
	Mixed pain	155	20.5	55	35.5	100	64.5	0.85	0.42
Tumor stage	Non-metastatic	256	33.8	166	64.8	90	35.1	Ref.	
	Metastatic	501	66.2	159	31.7	342	68.3	3.96	<0.001
Location of radiotherapy centers	North of Italy	178	23.5	57	32.0	121	68.0	Ref.	
	Center of Italy	75	9.9	47	62.7	28	37.3	0.28	<0.001
	South of Italy	504	66.6	221	43.8	283	56.2	0.60	<0.001

Legend: ECOG-PS: Eastern Cooperative Oncology Group Performance Status Scale; NRS: Numerical Rating Scale; OR: odds ratio. Percentages in “all patients” columns are column percentages. Percentages in “PMI” columns are row percentages.

**Table 3 cancers-16-00109-t003:** Multivariable analysis: odds ratios of adequate pain management (only 757 patients with pain or undergoing analgesic therapy included).

	OR	*p*-Value
Aim of treatment		
Curative	Ref.	
Palliative	2.52	<0.001
ECOG-PS		
1	Ref.	
2	1.63	0.017
3	2.23	0.002
4	5.31	0.009
Type of pain		
Cancer pain	Ref.	
Non-cancer pain	0.38	<0.001
Mixed pain	1.12	0.57
Location of the radiotherapy center		
North of Italy	Ref.	
Center of Italy	0.25	<0.001
South of Italy	0.42	<0.001

Legend: ECOG-PS: Eastern Cooperative Oncology Group Performance Status Scale; OR: odds ratio.

## Data Availability

Data supporting the reported results can be found at Radiotherapy Unit—A.G. Morganti of the IRCCS Azienda Ospedaliero-Universitaria di Bologna.

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
