# Peer review of "Adequacy of Pain Management in Patients Referred for Radiation Therapy: A Subanalysis of the Multicenter ARISE-1 Study"

_cancers, 2023, doi:10.3390/cancers16010109_

Round 1

Reviewer 1 Report

Comments and Suggestions for Authors

The authors of the manuscript “Adequacy of Pain Management in Patients Referred to Radiation Therapy: a Subanalysis of the Muliticenter ARISE-1 Study” provide an interesting view on pain, pain management and related factors in patients with cancer referred to radiation therapy in Italy. However before publication some issues have to be clarified.

The focus is on adequacy of pain management and associated factors, based on the PMI.  This manuscript is compact and well elaborated. However, whereas the MPMI is introduced and suggested to be superior to the PMI, main results are based on PMI. This is confusing. Moreover, further validation  of the MPMI is recommended, howevever  not added. See also below.

Introduction.

Line 88-94 No references about pain treatment in RT patient are provided.

Line 97 Please use superscript for ref 2.

Line 103 please use association instead of correlation (similar to association in line 105).

Materials and methods

Line 129-133 please add references for ECOG and NRS.

Line 143-148 MPMI. I suggest to split in primary and secondary endpoints. MPMI is clearly a secondary endpoint, based on limited results and absence of validation. Please make explicitly clear in methods and results that primary analyses are on the PMI, not on MPMI.

Results

Line 200 heading 3.1.2. Pain Management Index. This seems about the Modified PMI? Please change or explain.

Discussion

The authors do mention that the MPMI has not been validated yet. It would be very supportive if  a preliminary short validation, using the current data,  would be added in a supplementary file: e.g. differences between PMI and MPMI based on sex, age and other used parameters could already be assessed; regression analyses could also be performed using the MPMI.

Author Response

Comment 1:

The authors of the manuscript “Adequacy of Pain Management in Patients Referred to Radiation Therapy: a Subanalysis of the Muliticenter ARISE-1 Study” provide an interesting view on pain, pain management and related factors in patients with cancer referred to radiation therapy in Italy. However before publication some issues have to be clarified.

The focus is on adequacy of pain management and associated factors, based on the PMI.  This manuscript is compact and well elaborated. However, whereas the MPMI is introduced and suggested to be superior to the PMI, main results are based on PMI. This is confusing. Moreover, further validation  of the MPMI is recommended, howevever  not added. See also below.

Answer 1:

We sincerely thank you for your positive remarks about our manuscript, "Adequacy of Pain Management in Patients Referred to Radiation Therapy: a Subanalysis of the Multicenter ARISE-1 Study." We appreciate your observations regarding the use of the Pain Management Index (PMI).

In response to your comments, we agree that introducing an unvalidated index like the MPMI without prior validation within the study could be problematic. However, our intention was to address a known limitation of the PMI, particularly in cases where pain management appears adequate (as indicated by PMI scores), even in patients experiencing clinically significant pain.

Therefore, as a secondary objective, we distinguished two groups: Group A (patients with PMI > 0 and controlled pain, NRS 0-4) and Group B (patients with PMI < 0 or with uncontrolled pain, NRS > 4). This categorization aimed to evaluate not just the adequacy but also the overall effectiveness of pain therapy.

Based on your suggestion, we have revised the relevant paragraph to clarify this approach.

Here is the original paragraph: ”However, the PMI has known limitations. In particular, patients experiencing intense pain despite the use of strong opioid drugs may have a PMI value of 0, incorrectly indicating the adequacy of therapy. To address this shortcoming, we introduced a modified index called the Modified Pain Management Index (MPMI). The MPMI aimed to compensate for the limitations of the PMI. Patients with PMI < 0 or patients with PMI ≥ 0 but with a pain score > 1 (NRS > 4) were classified as having a negative MPMI score (MPMI-). Conversely, patients with PMI ≥ 0 and pain score < 2 (NRS ≤ 4) were classified as having a positive MPMI score (MPMI+).”

The revised paragraph now reads: “However, the PMI is known to have limitations. Notably, patients experiencing intense pain despite strong opioid medication may still have a PMI value ≥ 0, inaccurately suggesting adequate pain management. To address this discrepancy, our study introduces a secondary analysis that classifies patients into two distinct groups: Group A consists of patients with PMI < 0 or those with PMI ≥ 0 but experiencing substantial pain (NRS > 4), indicative of poorly managed or ineffective pain therapy; Group B includes patients with PMI ≥ 0 and a pain score ≤ 4, representing those with adequate and effective pain management.”.  [page 3-4]

Corresponding sections in the results and discussion have also been modified to align with this change in objectives.

We believe these modifications address your concerns and clarify the use of PMI and MPMI in our study.

Comment 2:

Introduction.

Line 88-94 No references about pain treatment in RT patient are provided.

Answer 2:

Thank you for your comment. According to your suggestion we added the references:

  1. Mitera, G.; Fairchild, A.; DeAngelis, C.; Emmenegger, U.; Zurawel-Balaura, L.; Zhang, L.; Bezjak, A.; Levin, W.; Mclean, M.; Zeiadin, N.; et al. A Multicenter Assessment of the Adequacy of Cancer Pain Treatment Using the Pain Management Index. J. Palliat. Med. 2010, 13, 589-93.
  2. Vuong, S.; Pulenzas, N.; DeAngelis, C.; Torabi, S.; Ahrari, S.; Tsao, M.; Danjoux, C.; Barnes, T.; Chow, E. Inadequate pain management in cancer patients attending an outpatient palliative radiotherapy clinic. Support. Care Cancer. 2016 Feb, 24, 887-892.

4.Raphaëlle Dantigny , Arnaud Tanty, Philippe Fourneret ,Nicolas Genin, Béatrice Bayet Papin, Mireille Mousseau, Ngoc-Hanh Hau Desbat. Prevalence of pain in radiotherapy and improvement of its management. Bull Cancer. 2018 Dec;105(12):1183-1192.

Comment 3:

Line 97 Please use superscript for ref 2.

Answer 3:

Thank you very much. Correction done.

Comment 4:

Line 103 please use association instead of correlation (similar to association in line 105).

Answer 4:

Thank you for your comment. We modified as suggested. [page 3]

Comment 5:

Materials and methods

Line 129-133 please add references for ECOG and NRS.

Answer 5:

Thank you for your comment. According to your suggestion we added the references:

  1. Oken MM, Creech RH, Tormey   DC et al. Toxicity and response criteria of the Eastern Cooperative Oncology Group. Am J Clin Oncol  1982;5:649–655.
  2. W.H. Oldenmenger, C.C. van der Rijt. Feasibility of assessing patients' acceptable pain in a randomized controlled trial on a patient pain education program. Palliat Med, 31 (6) (2017), pp. 553-558
  3. Downie WW, Leatham PA, Rhind VM, Wright V, Branco JA, Anderson JA. Studies with pain rating scales. Ann Rheum Dis 1978; 37: 378–81.

Comment 6:

Line 143-148 MPMI. I suggest to split in primary and secondary endpoints. MPMI is clearly a secondary endpoint, based on limited results and absence of validation. Please make explicitly clear in methods and results that primary analyses are on the PMI, not on MPMI.

Answer 6:

Thank you for your continued guidance. Following your feedback, we have removed the Modified Pain Management Index (MPMI) from our study. Instead, as per our previous response (Answer 1), we have implemented your suggestion by classifying patients into two groups: Group A (patients with PMI < 0 or those with PMI ≥ 0 but experiencing substantial pain, NRS > 4) and Group B (patients with PMI ≥ 0 and a pain score ≤ 4). This revised approach addresses the limitations of the PMI and aligns with our study objectives, without introducing a new, non-validated index. The manuscript has been updated accordingly in both the methods and results sections to reflect this change

Comment 7:

Results

Line 200 heading 3.1.2. Pain Management Index. This seems about the Modified PMI? Please change or explain.

Answer 7:

Thanks very much for the comment. Based on our responses 1 and 6, we have changed the title of the subsection to read: “analysis of patients with inadequately or ineffectively managed pain” [page 7]

Comment 8:

Discussion

The authors do mention that the MPMI has not been validated yet. It would be very supportive if  a preliminary short validation, using the current data,  would be added in a supplementary file: e.g. differences between PMI and MPMI based on sex, age and other used parameters could already be assessed; regression analyses could also be performed using the MPMI.

Answer 8:

Thank you very much for your valuable advice. As outlined in our previous responses, we have endeavored to improve our manuscript while avoiding the introduction of a non-validated index. We concur that the validation of a new index merits a dedicated study. Accordingly, we intend to pursue this in future research, where we can focus exclusively on the development and validation of a new index, providing it the comprehensive analysis it requires. We believe this approach will maintain the integrity of our current study while opening avenues for further exploration in subsequent work.

Reviewer 2 Report

Comments and Suggestions for Authors

Comments:

1. Additional patient information required: BMI, presence of cancer markers, and history of smoking and alcohol consumption.

2. Regarding metastasis: Can we specify the exact location(s) of metastasis?

3. Are there any cases of renal and/or bladder cancers among the patients?

4. Typographical errors noted: "Nord" should be corrected to "North" in both Table 1 and Table 3.

5. Considering the focus on pain management in this study, would it be more appropriate to measure pain intensity on a scale of 0-10 rather than using categories 0-3? The results can be recorded as numerical values from 0 to 10 for clarity.

Author Response

Comment 1:

Additional patient information required: BMI, presence of cancer markers, and history of smoking and alcohol consumption.

Answer 1:

Thank you for your insightful comments regarding the inclusion of additional patient information in our study. We agree that factors such as BMI, presence of cancer markers, and history of smoking and alcohol consumption, metastatic site and type of tumor are indeed significant in the context of pain management in radiotherapy patients.

However, we would like to clarify that our study is a secondary analysis of the ARISE-1 study, which was a multicenter project involving over 2000 patients. Due to the practical constraints of such a large-scale study, the number of data points collected for each patient was limited. We appreciate your suggestion and acknowledge that the inclusion of these factors could have provided additional valuable insights.

To address this, we have added a section in our discussion acknowledging the limitation of our dataset in this regard. We hope this addition provides a comprehensive understanding of our study context and its contributions to the field.

"Moreover, in reflecting upon the scope of our study, we acknowledge certain limitations in the dataset utilized. Notably, our analysis did not include factors such as BMI, presence of cancer markers, metastatic site, and history of smoking and alcohol consumption. These elements have a potential impact on pain management efficacy in radiotherapy patients. This limitation stems from the practical constraints encountered in the multicenter ARISE-1 study, our primary data source, which involved an extensive cohort of over 2000 patients. As such, our findings should be interpreted with consideration of these omitted variables, and future research may benefit from their inclusion to further elucidate the complexities of pain management in this context."  [discussion, page 11]

Comment 2:

Regarding metastasis: Can we specify the exact location(s) of metastasis?

Answer 2:

Thank you for your query about specifying the exact locations of metastasis in our study. We would like to refer you to our response to your previous comment (Answer 1), where we explained the scope and limitations of the data collected in the ARISE-1 study. Due to the practical constraints of this large-scale, multicenter study, the level of detail regarding metastasis locations was limited. We appreciate your understanding of these limitations and believe that the additional section added to our discussion adequately addresses this aspect of our study.

Comment 3:

Are there any cases of renal and/or bladder cancers among the patients?

Answer 3:

Thank you for your question regarding the inclusion of renal and/or bladder cancer cases in our study. In our dataset, these less frequent tumor types were grouped under the category “others”. This categorization was due to the relative rarity of these cancer types within the large cohort of the ARISE-1 study. We recognize that this classification may limit specific insights into these particular cancer types in the context of pain management in radiotherapy. We have noted this as a limitation in our discussion section to provide clarity on the scope of our study data.

Comment 4:

Typographical errors noted: "Nord" should be corrected to "North" in both Table 1 and Table 3.

Answer 4:

Thank you very much. Correction done.

Comment 5:

Considering the focus on pain management in this study, would it be more appropriate to measure pain intensity on a scale of 0-10 rather than using categories 0-3? The results can be recorded as numerical values from 0 to 10 for clarity.

Answer 5:

Thank you for your suggestion regarding the use of a 0-10 scale for measuring pain intensity. We agree that a more granular scale can provide detailed insights into pain levels. However, our study primary objective was to evaluate the Pain Management Index (PMI), which utilizes a 0-3 categorization system for pain. This categorization aligns with the standards set in the ARISE-1 study, from which our data was derived. Therefore, to maintain consistency with the PMI framework and the original study design, we adhered to the 0-3 pain categorization.

Round 2

Reviewer 1 Report

Comments and Suggestions for Authors

The authors have made great improvements in version two. Using the secondary analysis with the additional criterion for the PMI is clear, however the heading 3.1.2. 'Analysis of patients with inadequately or ineffectively managed pain'  remains somewhat  confusing: this is in the end similar to what the PMI in the primary analyses is about? Then, when reading the first section of the discussion, line 238-240, the distinction  (between PMI<0 and group A) is not very clear. Therefore, I would suggest the authors to reformulate once more.

Author Response

Thank you very much for your further comment. We agree that the text can be confusing. Therefore, to better clarify the meaning of these parts of the manuscript, we have modified the sentence:

“Considering all patients enrolled in the study, the rate of patients in Group A, those with PMI < 0 or with PMI ≥ 0 but experiencing substantial pain (NRS > 4), indicative of poorly managed or ineffective pain therapy was 53% (Figure 1).”

Of the results section as follow:

“Considering all patients enrolled in the study, the rate of patients in Group A, (PMI < 0 or with PMI ≥ 0 but experiencing substantial pain: NRS > 4), indicative of poorly managed or ineffective pain therapy was 53% (Figure 1).”

Moreover, we have changed the sentence: “Moreover, the rate of patients with pain inadequately or ineffectively managed was 73%.”

In the discission section as follows: “Moreover, the rate of patients with pain inadequately or ineffectively managed (group A: PMI <0 or PMI ≥0 but NRS >4) was 73%.”

Reviewer 2 Report

Comments and Suggestions for Authors

No more comments

Author Response

Thank you.